# Powerline right-of-way management and flower-visiting insects: How vegetation management can promote pollinator diversity

**Laura Russo** [1,2]*, **Hannah Stout**[3], **Dana Roberts**[4], **Bradley D. Ross**[5], **Carolyn G. Mahan**[5]

**1** Department of Ecology and Evolutionary Biology, University of Tennessee, Institute of Agriculture, Knoxville, TN, United States of America, **2** Biology Department, Penn State University, University Park, PA, United States of America, **3** Independent Researcher, State College, PA, United States of America, **4** Department of Entomology, Penn State University, University Park, PA, United States of America, **5** Department of Biology and Environmental Studies, Penn State Altoona, Altoona, PA, United States of America

* lrusso@utk.edu

**Data Availability Statement:** All relevant data are within the manuscript and its Supporting Information files. The raw data are available in Appendix 2.

## Abstract

Loss in the availability of early successional habitat is a threat to pollinator populations. Given that powerline rights-of-way (ROW) must be managed to maintain early successional habitat, preventing vegetation from interfering with electrical lines, they have the potential to provide conservation benefits for wild pollinators. Moreover, it is possible to provide conservation benefits with no additional cost to land managers. We surveyed flower-visiting insects over two years in different vegetation management treatments in a long-term research ROW to determine which best promoted pollinator abundance and species richness. We found that the ROW had stabilized in an early successional state soon after its establishment and that this early successional state could be maintained with low levels of periodic maintenance. We collected a high diversity of flower-visiting insects (126 bee species and 179 non-bee morphospecies) in six ROW plots. Higher levels of herbicide application had a negative effect on bee species richness, but low levels of herbicide application were compatible with a high abundance and species richness of flower-visiting insects, including several rare species. Moreover, this effect was seen only in the bee community, and not in non-bee flower-visiting insects. Our results suggest further research into the conservation value of ROW for pollinators is warranted. We demonstrate that there is substantial potential for pollinator conservation in ROW, compatible with low-cost vegetation management.

## Introduction

The loss of early successional habitat is a threat to biodiversity [1, 2]. For this reason, powerline rights-of-way (ROW), which are managed to maintain a permanent state of early succession [3], may provide valuable habitat for many species [4]. Indeed, ROW have high plant species richness [4], and provide habitat for bees [5, 6], butterflies [3], and birds [7]. ROW comprise a large amount of land in a corridor-like pattern of continuous lines, enhancing their potential to connect plant and animal populations [8]. Indeed, ROW comprise approximately 2–3

**Funding:** This research was funded by and in collaboration with Penn State Altoona, Frost Entomological Museum, Penn State Extension, DOW Agro Industries, Asplundh, First Energy, and PECO Energy. L. Russo was funded by NSF grant (DMS-1313115) and a Marie Curie Fellowship (FOMN-705287).

**Competing interests:** We received funding from commercial sources: DOW Agro Industries, Asplundh, First Energy, and PECO Energy. These funders did not place any restrictions on the publication of data. This funding does not alter our adherence to PLOS ONE policies on sharing data and materials.

million ha in the US and traverse a wide array of habitats and landscapes [9]. For example, in New York, electric utilities manage 9 times as much early successional habitat as all federal, state, and non-governmental organizations [7]. Many pollinator species are declining [10–12], but we know relatively little about these insects, despite the essential services they provide to agriculture [13] and ecosystems in general [14]. One of the main causes of decline in pollinator populations is the loss of habitat rich in flowering plants [11]. Thus, ROW have great conservation potential for populations of pollinators [9].

Because ROW are managed to keep vegetation below a certain height, there is the opportunity to find win-win scenarios for management and conservation. The vegetation management of ROW maintain the land immediately below powerlines in an unforested state, which has been shown to relate positively with bee diversity and abundance, particularly for solitary bees [15]. Moreover, naturally developed vegetation appears to have greater benefits for pollinating insects than planted crops [16]. Maintenance of early successional habitat simultaneously addresses electrical transmission safety concerns while, at no additional cost, achieving conservation objectives by providing habitat rich in flowering plants for pollinators [6]. For this reason, we evaluated flower-visiting insect diversity in the Vegetation Research and Demonstration Project at State Game Lands 33 (SGL33) in Centre County, PA. The study area has been surveyed continuously for sixty-four years to evaluate the effects of vegetation management on biodiversity [17, 18]. Previous studies at SGL33 have monitored plant [18], mammal [19], bird [20], reptile, amphibian, and butterfly diversity [21].

Our research objectives were to 1) assess species richness of flower-visiting bee and non-bee insects in the Vegetation Research and Demonstration Project at SGL33, 2) determine the effect of increasing herbicide application on flower-visiting insect abundance and species richness within a powerline ROW, and 3) address how long-term vegetation management affects the distribution of flower-visiting insects in a powerline ROW.

## Materials and methods

### Plots and data collection

The research plots at SGL33 (Central PA: 40.8512, -78.1422) stabilized into an early successional state over 50 years ago. Current management involves visiting plots every 4–5 years, mowing where necessary, and applying herbicide according to four current treatment categories: 1) hand cutting (no herbicide, HC), 2) low volume basal herbicide (~10 litres/ha, LVB), 3) low-volume foliar herbicide (<10 litres/ha, LVF), and 4) high volume foliar herbicide (~70–250 litres/ha, HVF) (S1 Table in S1 Appendix). The high-volume foliar herbicide was a water-based broad-leaf herbicide broadcast sprayed across vegetation. This treatment was non-selective and used large amounts of a mixture of Aminopyralid, Imazapyr, Triclopyr, Picloram, and Glyphosate (see S1 Table in S1 Appendix for concentrations). The low volume basal treatment was an application of an oil-based herbicide to the root collar and trunk of shrubs and small trees. This treatment used a mixture of Aminopyralid, Imazapyr, Triclopyr and was applied selectively directly to non-compatible woody vegetation (i.e. vegetation with the potential to interfere with the powerlines). This method was known as 'cut and squirt': cut the wood and apply the herbicide selectively to the cut. The low volume foliar herbicide applications selectively applied an ultra-low volume of an oil-based herbicide to leaves with a nozzle applicator. This treatment was still selective, but the entire plant sprayed rather than just the cut area. The method was to wet the leaves and entire non-compatible plant using a mixture of Glyphosate and Imazapyr.

The powerline ROW is owned and managed by FirstEnergy, which allowed us access to do our surveys. We also obtained permits from the PA Game Commission for site visits. We

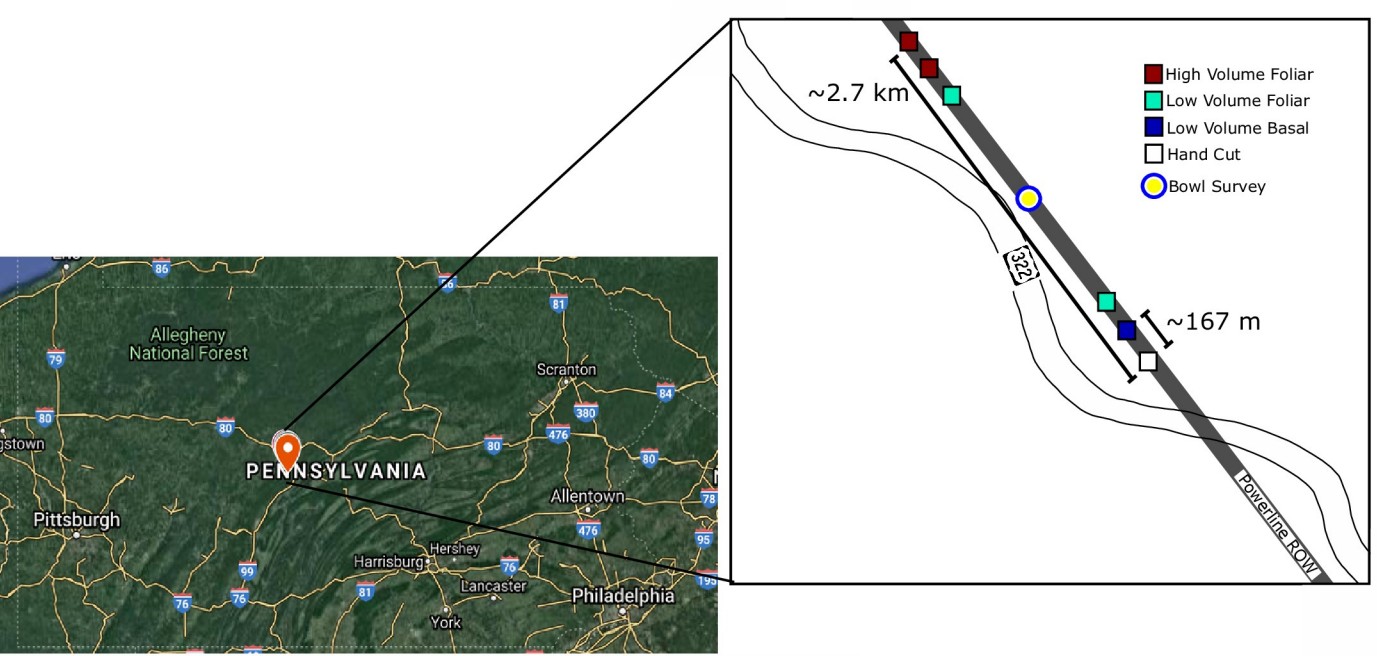

**Fig 1. A map of the study area within Pennsylvania and the distribution of the plots within the powerline ROW.**

surveyed six 50 m x 25 m plots at SGL33, where herbicide treatments had been applied four years before our study began in 2012, then again in the autumn of 2016 after our first summer of collection and prior to our second year of collection. Each of these plots was separated by at least 167m (greatest distance between two plots = 2.7 km, Fig 1). We surveyed two plots each of low volume foliar and high volume foliar, and one plot each of hand cut and low volume basal. However, we ultimately excluded the hand-cut treatments from the analysis of treatment effects because we felt that we were not able to accurately sample these plots due to overgrowth of thorny vegetation (see S1 Appendix for more details about hand cut treatments).

Establishing the baseline diversity of insect flower-visitors in the ROW has significant advantages for determining whether the ROW have conservation value, and moreover may serve as a comparison to future sampling efforts to determine whether insect populations in these ROW are stable. By comparing management strategies, on the other hand, we can identify win-win scenarios between management and conservation objectives for these ROW. Additionally, we can determine whether certain management strategies lead to lower pollinator abundance and diversity.

We conducted surveys by hand-netting insects visiting flowers on dry, warm (21 – 29C), and windless (<8kph) days between 10am– 4pm, conditions when most hymenopteran flower visitors actively forage [22]. In the summer of 2016, we conducted four sample periods, collecting at each plot twice a month (once in the morning and once in the afternoon) from May– August, and collected all insects found on flowers, as non-bee insects can be important pollinators [23]. We used plot-based sampling methods, where the collector covered the area of each treatment plot on foot during the sampling event, collecting all visible insects that contacted the reproductive parts of flowers within the plot. This sampling was not constrained to a transect within the plot and the collector was able to retrace their steps if they had additional time after covering the area of the plot once. In the summer of 2017, we conducted a similar sampling regime, but collected only bee specimens. Each sample period (in both years) involved 2

net-hours of sampling in the morning and 2 net-hours in the afternoon at each plot. Sixteen net-hours were spent at each plot—8 morning and 8 afternoon collections—across the season (96 hours each in 2016 and 2017, or 192 total hours). This sampling effort includes handling time for transferring specimens from the nets to collection jars. Ninety-six percent of bee specimens were identified to species with the help of Sam Droege (United States Geological Survey (USGS)) and the remaining 4% were identified to genus. From the 2016 collections, most non-bee specimens were identified to genus, species, or morphospecies with 55 identified to family and 4 to order [24–28]. Voucher specimens are stored at the Frost Entomological Museum at the Pennsylvania State University.

To better understand background diversity of bees in the ROW not found in our net collections, we also conducted one bowl survey in the ROW, but not in the treatment plots from July 26–27, 2016 [29]. The bowl survey was conducted in the approximate halfway point between the two farthest plots (Fig 1). They are included here to better describe the potential for ROW to be habitat for bees, as pan trapping sometimes collects species not found in net collections [30]. We placed 36 plastic bowls (blue, yellow, and white) containing 300mL water with unscented soap in a 30m x 25m plot in the ROW (S1 Fig in S1 Appendix) to provide a similar background plant community (Stout, unpub. data). These bowls were left out for 24 hours before being retrieved, and were placed on bare ground. Bowl sampling can act as a complement to hand-netting, as it attracts a distinct bee community, and various pollinator taxa prefer different bowl colors [31, 32]. However, bowl sampling is not reliable for sampling within small areas, so we did not deem it adequate for comparing the separate plots. Moreover, bowl sampling has been shown to collect only specific bee species, and is therefore not appropriate measure of background bee abundances [33, 34]. Specimens from the bowl survey were identified to better describe the baseline insect diversity in the ROW and described in the supplemental tables (S2 Table in S1 Appendix), but were not included in analyses as they were not collected within our treatment plots.

The plots are all surveyed for plant diversity once a year during peak bloom (July). These surveys involved determining the species richness of native flowering plants under 2 m in height compatible with ROW maintenance (e.g., forbs or plants with shrubby-growth form). Plant species within a 5-m radius plot placed in the center of each transect were identified and counted [35]. Although this annual survey does not constitute a complete measure of plant species richness across the year, we used these data here as a proxy for plant diversity in the 6 treatment plots.

### Data analysis

To determine whether management affected pollinator abundance and species richness on a per-sample basis (i.e. for each collection), we used general linear mixed effects models (GLMM) with the R package "lme4" [36]. These models handle uneven sampling and can partition variation to fixed and random effects. We used time of day and year as random effects. Then we tested for correlations between the following treatment variables: litres/hectare herbicide applied in 2012, litres/hectare herbicide applied in 2016, and a plant species richness survey conducted in July 2016 (S3 Table in S1 Appendix). We compared the management treatment as a categorical (treatment type) or continuous (litres of herbicide/hectare) as fixed effects, comparing model fit between these two ways of coding the management. We also used GLMMs to determine whether time of day or month of year had a significant effect on the abundance or richness of bees (2016 & 2017) or non-bees (2016) when we assigned the plot identity and year as random effects.

To test the hypothesis that the herbicide treatments would have a greater effect on less common bees, we repeated these analyses after removing the two most common bee species (*Apis*

*mellifera* and *Bombus impatiens*, together comprising 25% of the bee abundance) from the analysis.

For each GLMM that we built, we tested for overdispersion. Where there was overdispersion, we attempted a Laplace approximation, using a Poisson distribution. In many cases, these models were still overdispersed, and so we log-transformed the response variable (abundance or species richness), to alleviate the overdispersion where necessary.

To compare diversity in the different plots, we used a rarefaction analysis. Rarefaction analyses account for the relative abundances of insects in the different treatment plots by providing observed and estimated species diversity measures [37]. We used the "iNEXT" package in R [37] to standardize based on sampling effort and completeness. For the purposes of comparing the different treatments, we compare the asymptotic confidence intervals for species richness, Shannon diversity, and Simpson diversity in bees and non-bee insects with the maximum reference sample as our base sample size. Where these extrapolated confidence intervals do not overlap, there are significant differences at an alpha level of 0.05 [37].

Finally, to visualize whether insect community structure in treatment plots differed, we used non-metric multidimensional scaling (NMDS) [38]. This ordination method is used to visualise multivariate data; in this case, we use NMDS to analyse the rank orders of the abundance of each species found in each treatment plot to determine whether there were differences in community composition (i.e. differences in both the number and identity of flower-visiting species). The variation in the rank orders of all the different species are collapsed into two axes for visualisation purposes, and the polygons are drawn based on the environmental variables (treatment, time of day, and month). NMDS requires a distance matrix, for which we used a Bray-Curtis dissimilarity calculation. We used the package "vegan" in R for this analysis [38].

## Results

We net-collected 2,344 bee specimens representing 126 species over the course of the two study years (S4 Table in S1 Appendix). The most abundant species across the surveys was *Bombus impatiens*, which alone comprised 14.5% of the bee abundance, followed by *Apis mellifera* (10.5%) and *Ceratina dupla* (7.6%). All other bee species individually represented less than 5% of the sample. The bowl survey collected 36 bee specimens of 19 species (4 not found in the hand-netting survey, S2 Table in S1 Appendix). We found representatives of all six bee families present in North America, including one melittid (*Macropis ciliata*), and several other specialists, including *Colletes validus* (Colletidae), *Megachile pugnata* (Megachilidae), and *Melissodes trinodis* (Apidae). We also had two new state records for PA: *Heriades leavitti* (Megachilidae) and *Melissodes apicatus* (Apidae) (pers. comm. S. Droege, J. Ascher) (S4 Table in S1 Appendix). Compared to a recently published state list of the bee species of Pennsylvania (Kilpatrick et al 2020), we found roughly 28.8% (126 of 437) of the bee species of Pennsylvania in our powerline survey, including 34.8% (8 of 23) of the non-native bee species recorded in the state. Results including the abundance and species richness of bees in the hand-cut plots are in the supporting information (S2 Fig in S1 Appendix). The months did not differ significantly in bee abundance (P > 0.05), however, there was a significantly greater abundance of bees in the afternoon, relative to the morning (Table 1). There was also a significantly higher bee species richness in the afternoon, relative to the morning. In addition, July had a significantly higher bee species richness than May, although the other months did not differ significantly (Table 1).

The plant survey yielded a total of 49 species or morphospecies in the research sites (S3 Table in S1 Appendix). The basal low volume treatment had the highest total plant species

**Table 1. Results of GLMMs on the effects of time of day and month of collection on bee abundance and species richness.**

| Response Variable | Fixed Effect | Contrast | Random Effect | Effect Size | t value | p value |
|---|---|---|---|---:|---:|---:|
| Log(Abundance) | Time, Month | AM—PM | Year, Plot | **0.4** | **2.05** | **0.04** |
| | | AUG—JULY | | -0.01 | -0.04 | 0.97 |
| | | AUG—JUNE | | -0.14 | -0.52 | 0.61 |
| | | AUG—MAY | | 0.04 | 0.16 | 0.87 |
| | | MAY—JULY | | -0.05 | -0.19 | 0.85 |
| | | MAY—JUNE | | -0.18 | -0.68 | 0.5 |
| | | JULY—JUNE | | -0.13 | -0.46 | 0.63 |
| Log(Species Richness) | Time, Month | AM—PM | Year, Plot | **0.34** | **2.5** | **0.01** |
| | | AUG—JULY | | -0.2 | -1.03 | 0.3 |
| | | AUG—JUNE | | 0.02 | 0.1 | 0.92 |
| | | AUG—MAY | | 0.28 | 1.49 | 0.14 |
| | | MAY—JULY | | **-0.48** | **-2.48** | **0.01** |
| | | MAY—JUNE | | -0.26 | -1.39 | 0.16 |
| | | JULY—JUNE | | 0.22 | 1.13 | 0.26 |

Significant effects are bolded.

richness (30 plant species) as well as the highest number of plant species found only at that site (10 species). The lowest plant species richness was found at the two high volume foliar sites (16 and 17 species) and one of the low volume foliar sites (16 species). Eight plant species or genera were found at all sites: *Dennstaedtia punctilobula* (Hayscented fern), *Gaultheria procumbens* (Teaberry), *Hamamelis virginiana* (Witchhazel), *Lysimachia quadrifolia* (Whorled Loose-strife), Poaceae (grasses), *Pteridium* (Bracken ferns), *Rubus* (Blackberries), and *Solidago* (Gold-enrods). Oil-collecting bees of the genus *Macropis* are thought to specialize on *Lysimachia* species, so the presence of *Lysimachia quadrifolia* may help to explain the presence of *Macropis ciliata* (Melittidae) in the ROW. One *M. ciliata* was recorded in a high volume foliar and one in a low volume foliar site. *Colletes validus* (Colletidae) is a specialist on *Vaccinium* (Blueber-ries), which were recorded in 5 of 6 sites, though the bee was only collected in a high volume foliar site. *Megachile pugnata* and *Melissodes trinodis* specialize on Asteraceae; 6 of 7 Astera-ceae species in the plots were found only in the low volume basal and one of the low volume foliar sites and these are also where the bees were found. The seventh Asteraceae species (*Soli-dago*) was found in all sites. The bowl surveys collected other specialist bees for which we did not record the plant hosts (e.g. *Eucera pruinosa* and *Melissodes apicatus*) and it is possible they were attracted to the bowls as they were passing through the habitat.

## Bee response to herbicide treatments

There was a total of 78 sampling events at the 5 plots (excluding the hand-cut plot) across 2 years (2016 & 2017) and 2 times of day (morning and afternoon). Bee abundance and richness were correlated [39]) on a per sample basis (S5 Table in S1 Appendix). When using a Poisson distribution, the count data (response = abundance) were overdispersed, so we log-trans-formed the response. The variables of litres/hectare herbicide (treatments applied in 2016) and plant species richness (surveyed in July 2016) were significantly negatively correlated (S4 Table in S1 Appendix), and so could not be included in the same model. Plant species richness was the best overall predictor of bee abundance (Table 2). Models including the categorical treatments as a fixed effect were more parsimonious than the continuous litres/ha herbicide usage as a fixed effect. Because the herbicide applications were in 2016, we also tested for

**Table 2. Results of the GLMMs on the effect of treatment on bee abundance.**

| Response Variable | Fixed Effect | Contrast | Random Effect | Effect Size | t value | p value | AIC |
|---|---|---|---|---|---|---|---|
| Log(Abundance) | 2012 application (continuous) | NA | Year, Time | < 0.001 | -0.16 | 0.87 | 215.01 |
| | 2012 application (categorical) | HVF—LVF | Year, Time | -0.23 | -1.11 | 0.27 | 213.81 |
| | 2016 application (continuous) | NA | Year, Time | -0.001 | -1.34 | 0.18 | 213.24 |
| | 2016 application (categorical) | LVB—HVF | Year, Time | **0.57** | **2.14** | **0.03** | 213.16 |
| | | LVF—HVF | Year, Time | -0.04 | -0.18 | 0.86 | |
| | | LVF—LVB | Year, Time | **-0.61** | **-2.26** | **0.02** | |
| | **2016 plant species richness** | **NA** | **Year, Time** | **0.04** | **2.76** | **0.006** | **207.73** |

Significant effects and the most parsimonious model are bolded.

interactions between the year (2016 or 2017) and the categorical treatments, but there were no significant interactions (P > 0.05).

The low volume basal (LVB) herbicide treatment had a significantly higher log transformed bee abundance per sample than the high or low volume foliar, which did not differ from one another (Fig 2, Table 2). The higher abundance in LVB is probably driven by the most abundant bee species in our study plots. For example, *Bombus impatiens* was 3.24 and 4.28 times more abundant in LVB than in HVF and LVF, respectively. Indeed, when we removed the two most abundant bee species (*Apis mellifera* and *B. impatiens*) from the dataset and repeated the analysis, there was no significant effect of the herbicide treatments (S6 Table in S1 Appendix). Interestingly, when we removed *B. impatiens* and *A. mellifera*, the effect of plant species richness on bee abundance was also no longer significant, suggesting those abundant species were also driving the relationship between plant species richness and bee abundance, despite the fact the most abundant bee species were both generalist species.

Using a similar process for log-transformed bee species richness, models with plant species richness as a fixed effect and litres/hectare herbicide as a fixed effect (continuous) were similar, but both more parsimonious than the fixed effect of categorical treatment. Plant species richness was significantly positively associated with bee species richness (Table 3), while litres/hectare herbicide (treatments applied in 2016) was negatively associated with bee species richness (Table 3, Fig 3).

For bee species richness, we based our rarefaction analysis on the categorical herbicide applications (S7 Table in S1 Appendix). Low volume foliar had the highest Shannon and Simpson diversity indices; for these diversity indices, low volume basal also had a significantly lower value than the high volume foliar (Fig 4). The bee species collected in our surveys considered by expert opinion to be rare included: *Bombus fervidus*, *Bombus sandersoni*, *Macropis ciliata*, *Melissodes apicatus*, *Nomada xanthura*, and *Heriades leavitti*. Among these bees, there were not strong patterns in their abundance across the treatment plots. *Melissodes apicatus* was only found in a bowl sample, and the other five species were distributed across the treatment plots (S3 Table in S1 Appendix).

Our NMDS plots of bee communities showed overlap in the different treatment plots (Fig 5, S3A Fig in S1 Appendix). There was also substantial overlap in time and year (S3C, S3D Fig in S1 Appendix). However, months clustered into visibly different groups, suggesting the community composition of bee species changes over the summer (Fig 5, S3B Fig in S1 Appendix).

## Non-bee flower-visiting insect response to herbicide treatments

We net-collected 744 non-bee specimens representing 179 morphospecies (S8 Table in S1 Appendix). GBIF records of insect species in Pennsylvania include more than 9200 species,

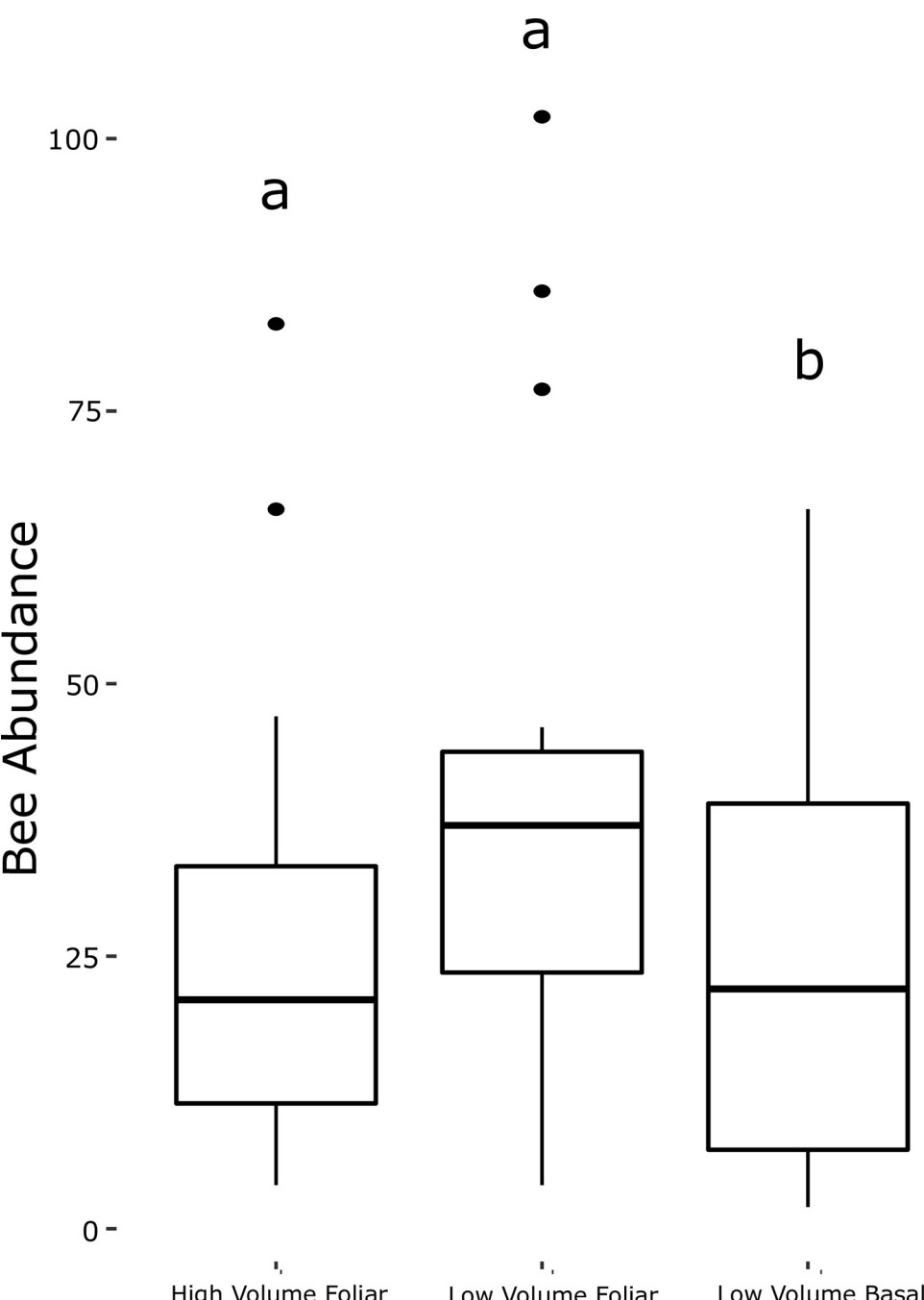

**Fig 2. Boxplots showing the relationship between three categorical treatment variables relating to the herbicide application at the ROW and the log-transformed bee abundance in each sample.** The boxplots represent the first to third quartile of the data (box), the median value (horizontal bar), with whiskers representing the quartiles ± 1.5 * the interquartile range. The outliers here are data points that lie beyond the range of the whiskers.

**Table 3. Results of GLMMs of the treatment effect on bee species richness.**

| Response Variable | Fixed Effect | Contrast | Random Effect | Effect Size | t value | p value | AIC |
|---|---|---|---|---|---|---|---|
| Log(Species Richness) | 2012 application (continuous) | NA | Year, Time | < 0.001 | -1.39 | 0.17 | 159.61 |
| | 2012 application (categorical) | HVF—LVF | Year, Time | 0.09 | 0.58 | 0.56 | 161.19 |
| | **2016 application (continuous)** | **NA** | **Year, Time** | **-0.002** | **-2.21** | **0.03** | **156.73** |
| | 2016 application (categorical) | LVB—HVF | Year, Time | -0.18 | -1.42 | 0.16 | 161.15 |
| | | LVF—HVF | Year, Time | -0.18 | -1.1 | 0.27 | |
| | | LVF—LVB | Year, Time | 0.1 | 0.5 | 0.62 | |
| | **2016 plant species richness** | **NA** | **Year, Time** | **0.03** | **2.34** | **0.02** | **156.18** |

Significant effects and the most parsimonious model are bolded.

however these include many species that would never be found on flowers. When we restrict the GBIF records to an example group of non-bee flower visitors, Syrphidae, our surveys found roughly 8% of the species in the state (13 of 165 species) [40]. The bowl survey collected 49 non-bee specimens of 27 morphospecies (15 not found in the net survey). Excluding the hand-cut plot, we conducted 39 collections, at two times of day (morning and afternoon) and across four months (May–August). Non-bee abundance correlated significantly with morphospecies richness. There was no correlation between non-bee abundance or richness and our proxy of plant richness (measured in July 2016) (S5 Table in S1 Appendix). Month of the year, but not time of day, was a strong predictor of non-bee abundance, where August had significantly more non-bee insects than any other month (S9 Table in S1 Appendix). In addition, June and July had significantly more non-bee insects than May, but did not differ from one another. There were similar effects for non-bee morphospecies richness (S9 Table in S1 Appendix).

With log-transformed non-bee abundance as a response variable and month of sampling as a random effect, the model with the lowest AIC value was one with the fixed effect of litres/ha herbicide (applied in 2016). However, the litres/ha herbicide was not a significant predictor of non-bee abundance (S10 Table in S1 Appendix). For non-bee morphospecies richness, the model with the lowest AIC was also one with the fixed effect of litres/ha herbicide, but it similarly was not a significant predictor of non-bee morphospecies richness (S10 Table in S1 Appendix).

## Discussion

### Species richness of flower-visiting bee and non-bee insects

We found approximately 29.7% (130 of 437) of the bee species of Pennsylvania in a relatively small powerline ROW, as well as 179 non-bee morphospecies of flower-visiting insects [41]. We also found rare species, such as *Macropis ciliata* (Melittidae), and new state records, *Heriades leavitti* (Megachilidae) and *Melissodes apicatus* (Apidae) (pers. comm. S. Droege, J. Ascher). The diversity in our samples is partially due to the fact that we sampled in both the morning and afternoon across the active season of pollinating insects. Other studies of bees in ROW have also found high species richness (163 species) and rare bee species [6]. It is therefore clear that ROW can be valuable habitat for pollinators, even for rare species. Powerline ROW represent corridors that connect diverse habitat and landscape types, and comprise approximately 2–3 million ha in the US [9]. Ecologically, powerline vegetation management keeps the plant community in early succession to prevent the establishment of tall woody species [3] and, once arrested into an early successional state, ROW can be managed with little

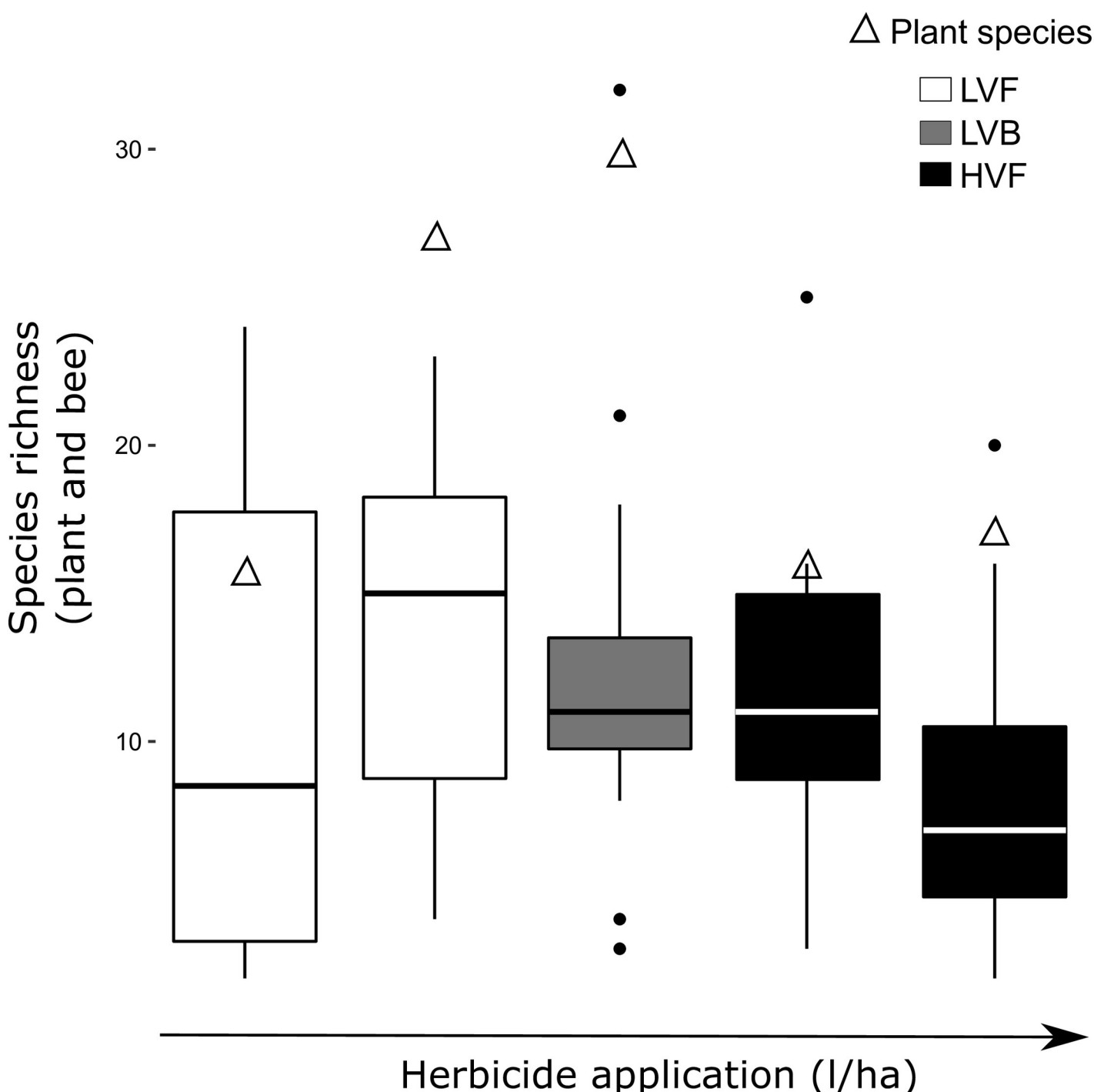

**Fig 3. Patterns of plant species richness (triangles), as surveyed in July 2016, and bee species richness in each of the plots in the ROW (boxplots and points).** The shading refers to the litres/ha herbicide applied at the treatment plots in 2016 (LVF, LVB, HVF), and the bars are arranged from lowest herbicide usage per site on the left, to highest on the right.

additional effort. At SGL33, land managers found that the establishment management treatments were no longer required after the plant communities stabilised and were able to adapt in their management of the ROW. For this reason, most of the plots we surveyed were maintained with small applications of herbicide once every 4–5 years.

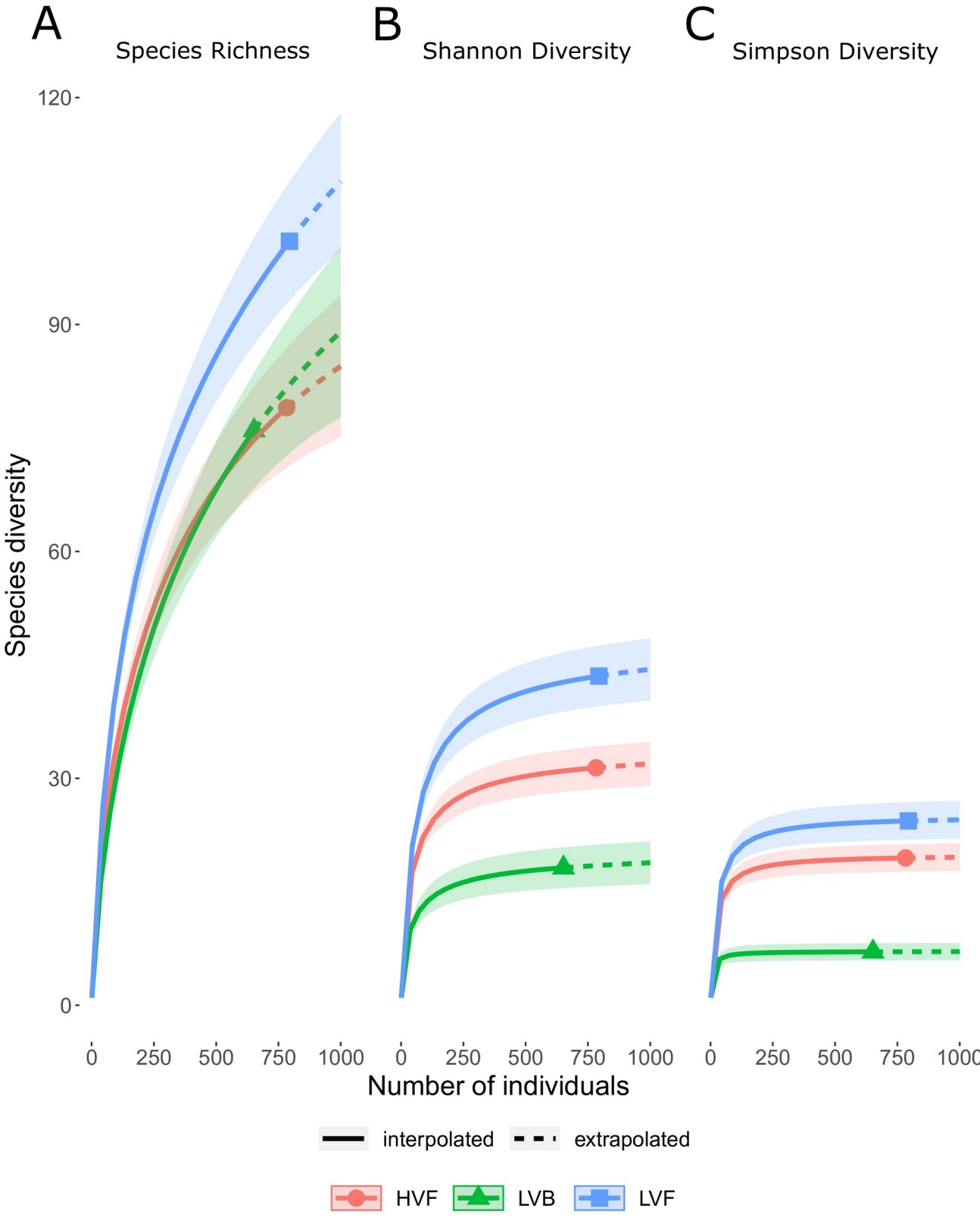

**Fig 4. Rarefaction analysis showing that the Low Volume Foliar (LVF) herbicide treatment has the highest species richness, Shannon, and Simpson diversities.** On the other hand, low volume basal (LVB) and high volume foliar (HVF) have similar species richness, but the LVB treatment has lower Shannon and Simpson diversities.

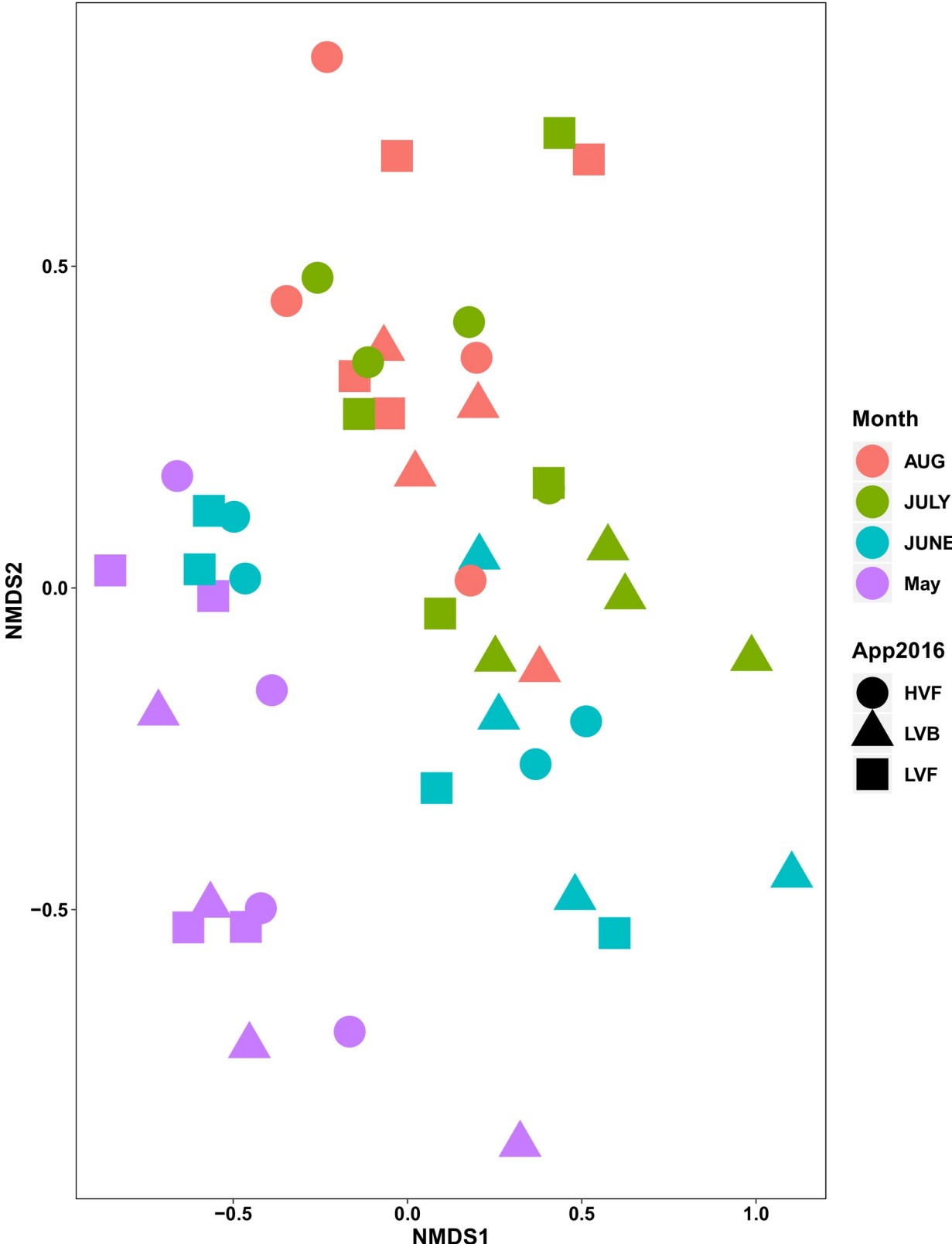

**Fig 5. An NMDS ordination plot showing separation of bee communities by month (colors) and herbicide treatments (shapes).**

## Effect of increasing herbicide usage on flower-visiting insect abundance and species richness

Our results showed that the strongest effect on bee abundance and species richness was driven by plant species richness in our study site, though lower, and more selective, levels of herbicide application had significantly higher abundances of common bee species, such as *Apis mellifera* and *Bombus impatiens*. It is possible that these species were more abundant in plots with lower herbicide application because of higher plant species richness. Increasing herbicide application also had a small negative effect on bee species richness.

Ideally, we would have compared these herbicide treatments to a hand-cut only treatment with no herbicide application, but the hand-cut plot was overgrown with briars and dense, thorny vegetation and more difficult to survey with hand nets. This meant that we were not able to fairly sample the flower-visiting insects of this plot. However, there was a good range of herbicide application in the other plots we sampled, from as low as 2.32 litres/ha (LVF) to as high as 249.66 litres/ha (HVF), and we did not see large differences in the abundance of bees between the two extremes of this large range. Instead, the low volume basal treatment (LVB) had significantly higher abundance than low volume foliar or high volume foliar, which did not differ significantly from one another (Fig 2). It is likely the observed higher abundance in LVB plots was driven by the relative abundance of the most common bee species in our study plot, *B. impatiens* and *A. mellifera*, and removing them from the analysis did eliminate the significant effects of both treatment and plant species richness. Indeed, the rarefaction analysis showed that the LVF treatment had the highest value for Shannon and Simpson diversities, while the LVB and HVF did not differ (Fig 4). The LVB treatment had a significantly lower Shannon and Simpson diversity, likely because of lower evenness due to the dominance of *B. impatiens* in samples at those plots.

## Effect of long-term vegetation management on the distribution of flower-visiting insects

In the NMDS plots, month of the year was the factor that lead to the largest separation of the bee communities, while there was substantial overlap in the bee communities among the different treatments. This suggests that the phenological trajectory of the community composition is stronger than the treatment effects. Indeed, because many bee species have short active periods, it is important to monitor plots across the season [42]. Furthermore, insect communities, and bee communities in particular, are highly variable from year to year [43]. Thus, ROW should be monitored for bee diversity continuously for several years before final management conclusions are drawn. Finally, the methods of herbicide application and the mixtures used differed between our different plots. In future research, it would be interesting to compare how herbicide formulations specifically affect the pollinator fauna in the ROW.

Plant species richness was the best predictor of bee abundance, but the best model for the treatment effect was the categorical treatment applications, which did not have a clear, direct effect on bee abundance. The plant species also corresponded well with the distribution of specialist bee species in the plots. For example, the aster specialist *Melissodes trinodis* was found in the plot with the highest Asteraceae species richness. These specialist bee species did not appear to be more sensitive to herbicide usage, and were present in the high volume application sites. On the other hand, there was a significant negative correlation between the litres/ha herbicide application and overall bee and plant species richness. Out of 130 bee species, 36 were represented by a single specimen across all our sampling. The distribution of these singletons in the treatment plots showed that 53% of singletons were found in LVF, 28% in LVB, and 17% in HVF plots. This may suggest less common bee species were more sensitive to the

increasing herbicide applications, possibly because the herbicides may remove less common plant species [44]. It is relatively difficult to rank bee species abundance, as the population sizes of most bees is unknown [45]. However, we had six bee species that were considered rare by expert opinion and there were no clear patterns in the distribution of these rare bee species among the treatment plots.

There was no effect of treatment on the abundance or diversity of non-bee flower-visiting insects on a per sample basis, but their abundance was affected by month (highest in August). Because we were focused on flower-visiting insects, this should only be considered a superficial survey of non-bee insect groups, some of which may visit flowers inconsistently. Most of the non-bee insects we collected were visiting flowers for nectar, or consumed pollen without acting as pollinators [46], and are included here because some can also be effective pollinators [23].

## Conclusions

On the basis of these surveys, judicious usage of herbicides is not necessarily detrimental to the abundance of pollinating insects, especially when used in low volumes and in a selective manner. In other words, we did not see a consistent negative effect of the herbicide applied at these ROW plots on bee abundance, although there was a significant negative effect of increasing litres/ha herbicide application on bee species richness, and a negative correlation between plant species richness and herbicide application on a per plot basis. Perhaps more importantly, our study shows without any additional cost or effort to land managers, maintaining ROW in an early successional state can result in a high diversity of flower-visiting insects. Across the six survey plots and in two summers, we found 126 bee species and 179 non-bee morphospecies. These plots also had representatives of all six bee families of North America, including several specialists and two new state records. These findings encourage continued monitoring of ROW to determine optimal management recommendations for pollinating insects.

## Supporting information

**S1 Appendix.**
(DOCX)

**S2 Appendix.**
(XLSX)

## Acknowledgments

We thank M. Herald, J. Berger, D. Keefer, C. Engstrom, and Z. Holden for collection and preparation, S. Droege for bee identification, A. Deans, I. Miko, G. Felton, K. Hoover, and M. Saunders for help with logistics and identification. All specimens are curated at the Frost Entomological Museum, University Park, PA.

## Author Contributions

**Conceptualization:** Hannah Stout, Bradley D. Ross, Carolyn G. Mahan.

**Data curation:** Laura Russo, Hannah Stout, Dana Roberts.

**Formal analysis:** Laura Russo.

**Funding acquisition:** Bradley D. Ross, Carolyn G. Mahan.

**Investigation:** Laura Russo, Dana Roberts, Bradley D. Ross, Carolyn G. Mahan.

**Methodology:** Laura Russo, Hannah Stout, Carolyn G. Mahan.

**Project administration:** Laura Russo, Hannah Stout, Dana Roberts, Bradley D. Ross, Carolyn G. Mahan.

**Resources:** Bradley D. Ross, Carolyn G. Mahan.

**Supervision:** Laura Russo, Hannah Stout, Carolyn G. Mahan.

**Validation:** Laura Russo.

**Writing – original draft:** Laura Russo.

**Writing – review & editing:** Laura Russo, Hannah Stout, Dana Roberts, Bradley D. Ross, Carolyn G. Mahan.

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
