## [Decision Letter · Decision Letter 0]

30 Jun 2020

PONE-D-20-14623

Powerline right-of-way management and flower-visiting insects: How vegetation management can promote pollinator diversity

PLOS ONE

Dear Dr. Russo,

Thank you and your co-authors for submitting your manuscript to PLoS ONE. To adjudicate your manuscript, three reviewers were solicited, all three accepted the solicitation, and returned their reviews with great rapidity. Given the incredibly abnormal and increasingly absurd 2020 that we are enduring, I hope you’ll join me in commending the reviewers in their thoughtful and expedient adjudication of this manuscript.

Despite differences in opinion with respect to a decision, with two arguing for minor revisions, one for rejection, the reviewer’s recommendations are largely consistent, and concurrent with my evaluation of this manuscript. Specifically, there is agreement that there is value to both these hard-won data and this research, and a desire to see it published. However, there are also consistent concerns both with the experimental design (i.e. lack of sufficient replication and attendant low statistical power) and a lack of clarity in many key details. The reviewers have graciously provided substantial and substantive comments and suggestions around these issues. And while two reviewers propose a decision of “minor revisions,” their comments, when combined with the third reviewer’s perspectives as well as my own impressions of the manuscript,argue for major revisions.

Please carefully consider the reviewer comments; I think they provide a roadmap to an improved manuscript. Specifically, 1) consider providing additional clarity in the materials and methods to address perceived opacity, 2) address the issue of small sample size head-on by underscoring the limitations (and, by extension, not overinterpreting the results), and 3) consider a comprehensive revision to provide a more cohesive narrative.  I look forward to seeing a revised version of this manuscript in the near future.

We look forward to receiving your revised manuscript.

Kind regards,

Mark A. Davis, Ph.D.

Academic Editor

PLOS ONE

Journal Requirements:

2. In your Methods section, please provide additional location information of the STUDY SITES, including geographic coordinates for the data set if available.

4.  Thank you for stating the following in the Financial Disclosure section:

"This research was funded by and in collaboration with Penn State Altoona, Frost Entomological Museum, Penn State Extension, DOW Agro Industries, Asplundh, First Energy, and PECO Energy. L. Russo was funded by NSF grant (DMS-1313115) and a Marie Curie Fellowship (FOMN-705287)."

We note that you received funding from commercial sources: DOW Agro Industries, Asplundh, First Energy, and PECO Energy.

Reviewers' comments:

Reviewer's Responses to Questions

**Comments to the Author**

1. Is the manuscript technically sound, and do the data support the conclusions?

Reviewer #1: Yes

Reviewer #2: No

Reviewer #3: Yes

2. Has the statistical analysis been performed appropriately and rigorously? 

Reviewer #1: Yes

Reviewer #2: No

Reviewer #3: Yes

3. Have the authors made all data underlying the findings in their manuscript fully available?

Reviewer #1: No

Reviewer #2: No

Reviewer #3: No

4. Is the manuscript presented in an intelligible fashion and written in standard English?

Reviewer #1: Yes

Reviewer #2: Yes

Reviewer #3: Yes

5. Review Comments to the Author

Reviewer #1: Studies on best management practices of ROWs are an often-overlooked avenue of research and these land resources can be an important resource and corridor for many beneficial insects. High-quality identification of bee species like those in this study are wonderful to see – they can be a very difficult group to work on. It was also positive to see the addition of new state records and rare species in these spaces. This is a nice, straight-forward study, but I have several questions and comments on methodology that I hope can be clarified/detailed in the manuscript.

LN67 – Change word order: “low volume basal herbicide”

LN68 – Can you briefly describe the difference between basal and foliar herbicides? For example, I assume basal herbicide application was targeted, but I’m not sure how that compares to low- or high-volume foliar herbicide (broadcast spray?). Do you know which herbicides were used? Do you know if the amounts of active ingredients (concentrations) were consistent across herbicide treatments?

LN81 – Was handling time included or excluded in terms of the two hours?

LN86 – I believe it should be Sam Droege, not Samuel, as I do not think I have seen him use Samuel in his publications.

LN91 – How long were bowls left out (I assume 24 hours)? Were they placed directly on the ground or mounted above ground or above the vegetation level?

LN103 – What is the surrounding vegetation type? How far apart are each of the plots? How large are each of the plots?

LN142 and LN170 – Perhaps one of the reasons your bowl survey abundance was low was because you mentioned that Bombus impatiens was one of the most common species you encountered and bumble bees are not caught frequently in bowls due to their large size (and might be more easily visually spotted for net collection).

LN146 and 216 – Don’t know what the family of Heriades leavitti or Macropis ciliata is off the top of my head -- would be more consistent to label all families.

LN229 – 231 – What herbicides were used? How soon after the application (month?) was your study started? Perhaps Bombus is more susceptible to sublethal effects of foliar herbicides (or their adjuvants)? There are a few studies showing impacts of some herbicides on honey bees and synergistic effects when exposed with other xenobiotics. Below are just a few references, but unfortunately most of these studies are confined to work on honey bees and only rarely are there studies on synthetic xenobiotics in Bombus. It would be very interesting if you could add herbicide identity as a factor in your model (but not necessary for this paper).

https://doi.org/10.1073/pnas.1803880115

https://doi.org/10.1242/jeb.117291

https://doi.org/10.1242/jeb.109520

https://doi.org/10.1038/srep40499

TableS3 – Table header indicates that numbers are “per ROW site”, it looks like these are numbers summed for each year. If I understand the requirements of PLOS Data Policy correctly -- with ROW site information it would allow for all of the underlying data to be present.

Reviewer #2: Russo et al., present a comparison of flower-visiting bee and non-bee communities across 5 rights of way plots with 3 different herbicide treatments. The treatment plot with the highest herbicide application had decreased bee richness, but non-bee insect richness did not differ by herbicide treatments. Bee abundance differed between herbicide treatments, being higher in the “low volume basal” plot, but abundance of non-bees did not differ. NMDS suggested a stronger seasonal signal in flower-visiting insect communities than treatment plot. The authors conclude that rights of way may harbor high diversity of flower-visiting insects, and herbicide application does not negatively affect bee abundance.

The resulting species list from sampling ROW habitats is valuable, as is the addition of insects to the state list. However, publication in PLoS One warrants a more robust study design, and more clearly presented objectives, methods, and results than the authors have included in their submission.

My major comments are:

1. Objective 1 is better suited for a different audience. The authors state as the first of 2 objectives (line 75): “…establish baseline diversity of insect flower visitors in the ROW.” It is not clear what “baseline diversity” is, nor the significance of it for the study. Perhaps the authors wish to document or describe the species collected in ROW plots. The effort toward collecting and identifying over 3000 insects is admirable and the resulting species list is a strength of the manuscript. However, this information may be more appropriate for a journal focused on regional faunas or descriptive data, and I encourage the authors to consider submission to such a venue.

2. I do not think the design is sufficiently robust to make conclusions on the effects of herbicide application on abundance and richness of insect communities. To address the second objective (“…compare the current management strategies to determine which has the potential to best support long-term pollinator diversity”), the abundance and diversity of flower-visiting insects are analyzed across 5 plots of 3 treatment types. I am sympathetic to the challenges inherent in large survey efforts of insects which often lead to smaller sample sizes due to trade-offs in how sampling is stratified across space and time. Nevertheless, the authors do not provide much description or context on the different plots beyond a categorical name and a liter/hectare amount of herbicide application. The significance of “basal” vs “foliar” is not explained (though I am assuming it has to do with how the herbicide was applied). Furthermore, although the study is cast as a comparison of management strategies, we are not provided background or context on these strategies in terms of how or why they are applied.

Below I have included some general comments by section.

Intro: The introduction is a good foundation yet very short and missing details which would enhance the rationale of the study. For example, more background context on powerline rights-of-way, how prevalent or widespread they are, the significance of early successional habitat for bees and other flower-visiting insects would enhance the study set-up. The addition of the research questions in this section would also improve the Introduction.

Materials and Methods: The first paragraph discusses treatment types by name, but it is not clear to me the significance of “high volume basal,” etc.

How were plots distributed within the ROW? Were they separated? Distances between them?

Did you cover the same extent of each plot or walk transects?

Were multiple people collecting in the same plot at the same time?

How did you select which flowers to target for sampling?

Each plot was sampled 4 times May through August- was that once per month?

More detail on the pan traps would be helpful, for example the purpose of multiple colors (for a non-entomological readership). Pan traps were placed between 2 plots- which ones and why? Why not place these within the net-sampled plots to potentially collect additional taxa? Line 97 states they were placed in an area with a similar background plant community, but it is not clear which plot or plot plant communities it is similar to.

Results: In general, the Results section is hard to follow and does not flow well.

Summary tables comparing richness and abundance values across plots and of the model results would be helpful additions to the manuscript.

I do not think figure captions were included with the submission.

Figure 2 is very confusing.

Discussion: Some of the beginning Discussion information would be helpful in the Introduction. Highlight the main findings of the study early on in this section. Additional supporting literature would help put the results of this study in context of what others have found. The argument is made that ROW harbor diversity of these groups, but how do the richness and abundance values per plot compare to other studies? It can be difficult to compare across studies due to sampling effort, however the reader has no metric for assessing high diversity or abundance.

Reviewer #3: Review Russo et al. Power line ROWs and pollinator diversity

This study attempts to draw some general inferences about the diversity of bee and non-bee pollinators visiting flowers in managed experimental ROW plots. Although the experimental design has very little replication, and the explanatory variable (liters/hectare of applied herbicide) is not populated evenly across the range of variation, there are still valuable insights from this work that could be useful to other researchers.

This paper could be strengthened by a analysis of the conservation value of each taxon (e.g. Natureserv G or S rankings) as another response variable to compare among the treatments. The interesting fact that several new distributional records were attained during this study speaks to knowledge gaps in the distribution of bee species in Pennsylvania; elaborating on these results in order to put these collections into regional or state context would improve the utility of this study. 126 bee species were observed, but how many were expected? This is a difficult question to answer and I’m not suggesting the authors should spend a great deal of time coming up with and justifying an expected species richness, but I am saying that a brief framing of these results in the context of the state and/or regional and local species pools would be useful pieces of information.

Although the study infers that herbicide treatments have effects on the plant community, which then drive differences among the pollinator community visiting the flower assemblages during sampling periods, differences in the similarity and composition of the plant communities themselves are not explored. Plant species richness is proposed as a correlate of bee species richness, but the identity of plant species that are common and privately exclusive to the treatments is not discussed. Some mention of pollinator preference or specificity is given as an ad hoc hypothesis explaining results (line 250) but these data could be used to test such a hypothesis.

This testable hypothesis is that (line 250) is that rare bee species might be more sensitive to increasing herbicide applications because herbicides may remove less common plant species. As mentioned above, no analysis of plant assemblages was provided but this hypothesis could at least be supported indirectly by evidence that plant species rare at low volumes herbicide applications are the species that are dropping out of the assemblage at higher volumes.

Another testable hypothesis, which is raised (line 171) but not tested, is that the significant difference in LVB bee abundance (wrt LVF and HVF) is a function of Bombus impatiens. A simple way to demonstrate that this species is driving this pattern is to remove this species from the analysis and compare abundance among plots.

Below I have some specific comments given with line numbers.

Line 98-100: The bees from the bowl surveys are presented without analysis, but it would be very interesting to know if these bees were more rare (higher imperilment ranking) than bees from the treatment plots. Additionally, the above comments about putting these collections into context with some treatment of the regional or state level fauna would really add meaning to these numbers.

Lines 143- 147: See above, discussion of particular taxa devoid of ancillary data on species conservation value or life history could be enhanced

Line 157: This result cites Figure 1 incorrectly (bee abundance and richness correlation is given in Table S4 but not Figure 1)

Line 163: There are several instances (see also 174 and 181) of the claim “significantly lower AIC”, this is misleading at best. The use of “significant” here may imply to a naïve reader that there is a hypothesis test comparing the AIC values, but that would be incorrect. I recommend that this language be modified perhaps as “Models including the categorical treatments as a fixed effect were more parsimonious than the continuous liters/ha herbicide usage as a fixed effect models”.

Line 167: A table of parameter estimates, confidence intervals and AIC scores would be very helpful to guide the reader through this section.

Line 176: cite of Figure 2 at end of Line 178 is in the wrong place, should be cited at the end of statistics in Line 176

Line 181: Table S6 indicates that the confidence intervals for LVF overlap with each of the other treatments.

Line 189: Again the discussion of taxonomic diversity would be greatly enhanced with some analysis or representation of the local and regional species pool. Data on how many species were collected are hard to place into the proper context without some idea of how many bees (or non-bees) would be expected from such surveys, or how many exist in the region as a hole.

Line 238: the phrasing here is awkward and might be improved by striking “bee community more changeable over time” and replacing with something like “the phenological trajectory is stronger than treatment effects.

Line 244- 247: Perfect opportunity to discuss ordination and analysis of plant assemblage data.

Line 249: This passage is unclear, whether singletons are singletons from the ENTIRE dataset of observations or singletons from a SITE or treatment. How many singletons were in each, and total?

6. PLOS authors have the option to publish the peer review history of their article (what does this mean?). If published, this will include your full peer review and any attached files.

Reviewer #1: No

Reviewer #2: No

Reviewer #3: No

---

## [Author Response · Author response to Decision Letter 0]

30 Sep 2020

PLOS ONE <em@editorialmanager.com>

Tue, Jun 30, 1:23 PM

to Laura

[External Email]

PONE-D-20-14623

Powerline right-of-way management and flower-visiting insects: How vegetation management can promote pollinator diversity

PLOS ONE

Dear Dr. Russo,

Thank you and your co-authors for submitting your manuscript to PLoS ONE. To adjudicate your manuscript, three reviewers were solicited, all three accepted the solicitation, and returned their reviews with great rapidity. Given the incredibly abnormal and increasingly absurd 2020 that we are enduring, I hope you’ll join me in commending the reviewers in their thoughtful and expedient adjudication of this manuscript.

Despite differences in opinion with respect to a decision, with two arguing for minor revisions, one for rejection, the reviewer’s recommendations are largely consistent, and concurrent with my evaluation of this manuscript. Specifically, there is agreement that there is value to both these hard-won data and this research, and a desire to see it published. However, there are also consistent concerns both with the experimental design (i.e. lack of sufficient replication and attendant low statistical power) and a lack of clarity in many key details. The reviewers have graciously provided substantial and substantive comments and suggestions around these issues. And while two reviewers propose a decision of “minor revisions,” their comments, when combined with the third reviewer’s perspectives as well as my own impressions of the manuscript, argue for major revisions.

Please carefully consider the reviewer comments; I think they provide a roadmap to an improved manuscript. Specifically, 1) consider providing additional clarity in the materials and methods to address perceived opacity, 2) address the issue of small sample size head-on by underscoring the limitations (and, by extension, not overinterpreting the results), and 3) consider a comprehensive revision to provide a more cohesive narrative. I look forward to seeing a revised version of this manuscript in the near future.

We look forward to receiving your revised manuscript.

Kind regards,

Mark A. Davis, Ph.D.

Academic Editor

PLOS ONE

Journal Requirements:

***We have corrected the style according to the style requirements.***

2. In your Methods section, please provide additional location information of the STUDY SITES, including geographic coordinates for the data set if available.

***We now include the latitude and longitude of the study site on line 64 and an additional supplemental figure with a map of the study sites within the context of PA.***

***FirstEnergy (one of our funding partners) has a right-of-way across the State Games Lands where our research site occurs. We also obtained a site visit permit from PA Game Commission. Penn State has a long-established research partnership with other state agencies in Pennsylvania, including the Pennsylvania Game Commission. We explain this on lines 93-96 of the methods and now include a statement in the acknowledgment section. “Permission to conduct research on site was granted by First Energy, which has right-of-way access across the State Games Lands, and the PA Game Commission, which issued a permit for site visits.“*** 

4. Thank you for stating the following in the Financial Disclosure section:

"This research was funded by and in collaboration with Penn State Altoona, Frost Entomological Museum, Penn State Extension, DOW Agro Industries, Asplundh, First Energy, and PECO Energy. L. Russo was funded by NSF grant (DMS-1313115) and a Marie Curie Fellowship (FOMN-705287)."

We note that you received funding from commercial sources: DOW Agro Industries, Asplundh, First Energy, and PECO Energy.

***We amended our competing interests statement to the following: “We received funding from commercial sources: DOW Agro Industries, Asplundh, First Energy, and PECO Energy. These funders did not place any restrictions on the publication of data. This funding does not alter our adherence to PLOS ONE policies on sharing data and materials.” We have included this competing interests statement in our amended cover letter.***

***We have updated references to the supporting information throughout the text and include captions for the supporting information at the end of the manuscript. We also include a second appendix with the raw data.***

5. Review Comments to the Author

Reviewer #1: Studies on best management practices of ROWs are an often-overlooked avenue of research and these land resources can be an important resource and corridor for many beneficial insects. High-quality identification of bee species like those in this study are wonderful to see – they can be a very difficult group to work on. It was also positive to see the addition of new state records and rare species in these spaces. This is a nice, straight-forward study, but I have several questions and comments on methodology that I hope can be clarified/detailed in the manuscript.

***We would like to thank the reviewer for your thoughtful and helpful comments. We have responded to them on a point-by-point basis below, demarcated by three asterisks (***). We refer to line number on the revised ms using the “Simple Markup” function in Microsoft word.***

LN67 – Change word order: “low volume basal herbicide”

***Changed as suggested.***

LN68 – Can you briefly describe the difference between basal and foliar herbicides? For example, I assume basal herbicide application was targeted, but I’m not sure how that compares to low- or high-volume foliar herbicide (broadcast spray?). Do you know which herbicides were used? Do you know if the amounts of active ingredients (concentrations) were consistent across herbicide treatments?

***We added more description here about the methods of application and the types of herbicide included in each on lines 78-90 and in table S1, along with the concentrations of the different herbicides.***

LN81 – Was handling time included or excluded in terms of the two hours?

***We now specify that handling time was included on lines 131-132.***

LN86 – I believe it should be Sam Droege, not Samuel, as I do not think I have seen him use Samuel in his publications.

***Changed as suggested.***

LN91 – How long were bowls left out (I assume 24 hours)? Were they placed directly on the ground or mounted above ground or above the vegetation level?

***We now specify that the bowls were left out for 24 hours and placed directly on the ground on line 145.***

LN103 – What is the surrounding vegetation type? How far apart are each of the plots? How large are each of the plots?

***We added more details about the size and separation of the plots on lines 93-96, and include a new supplemental figure showing the plot layout Fig. S1.***

LN142 and LN170 – Perhaps one of the reasons your bowl survey abundance was low was because you mentioned that Bombus impatiens was one of the most common species you encountered and bumble bees are not caught frequently in bowls due to their large size (and might be more easily visually spotted for net collection).

***We now cite a paper by Portman et al 2020 suggesting the bias of bowl-trapping on lines 146-151.***

LN146 and 216 – Don’t know what the family of Heriades leavitti or Macropis ciliata is off the top of my head -- would be more consistent to label all families.

***We now add the families for these bee species.***

LN229 – 231 – What herbicides were used? How soon after the application (month?) was your study started? Perhaps Bombus is more susceptible to sublethal effects of foliar herbicides (or their adjuvants)? There are a few studies showing impacts of some herbicides on honey bees and synergistic effects when exposed with other xenobiotics. Below are just a few references, but unfortunately most of these studies are confined to work on honey bees and only rarely are there studies on synthetic xenobiotics in Bombus. It would be very interesting if you could add herbicide identity as a factor in your model (but not necessary for this paper).

https://doi.org/10.1073/pnas.1803880115

https://doi.org/10.1242/jeb.117291

https://doi.org/10.1242/jeb.109520

https://doi.org/10.1038/srep40499

***We now include a description of the herbicides used. Unfortunately, we do not have the statistical power to test for the effects of individual herbicide mixtures on the bees, but we list this as a potential avenue for future research on lines 386-389.***

TableS3 – Table header indicates that numbers are “per ROW site”, it looks like these are numbers summed for each year. If I understand the requirements of PLOS Data Policy correctly -- with ROW site information it would allow for all of the underlying data to be present.

***We now include the raw data as an excel file in appendix 2.***

Reviewer #2: Russo et al., present a comparison of flower-visiting bee and non-bee communities across 5 rights of way plots with 3 different herbicide treatments. The treatment plot with the highest herbicide application had decreased bee richness, but non-bee insect richness did not differ by herbicide treatments. Bee abundance differed between herbicide treatments, being higher in the “low volume basal” plot, but abundance of non-bees did not differ. NMDS suggested a stronger seasonal signal in flower-visiting insect communities than treatment plot. The authors conclude that rights of way may harbor high diversity of flower-visiting insects, and herbicide application does not negatively affect bee abundance.

The resulting species list from sampling ROW habitats is valuable, as is the addition of insects to the state list. However, publication in PLoS One warrants a more robust study design, and more clearly presented objectives, methods, and results than the authors have included in their submission.

***We thank the reviewer for their helpful comments and respond to them on a point-by-point basis below, demarcated by three asterisks (***). Our line numbers refer to the revised manuscript in “Simple Markup” mode in Microsoft word.***

My major comments are:

1. Objective 1 is better suited for a different audience. The authors state as the first of 2 objectives (line 75): “…establish baseline diversity of insect flower visitors in the ROW.” It is not clear what “baseline diversity” is, nor the significance of it for the study. Perhaps the authors wish to document or describe the species collected in ROW plots. The effort toward collecting and identifying over 3000 insects is admirable and the resulting species list is a strength of the manuscript. However, this information may be more appropriate for a journal focused on regional faunas or descriptive data, and I encourage the authors to consider submission to such a venue.

***Thank you for your comment. The editor has given us a chance to revise and resubmit our manuscript for consideration in PLoS. We feel that the diversity of flower-visiting insects in powerline rights of way is of interest to a broad audience because 1) it establishes a record off which we can compare future sampling efforts to determine whether insect populations are changing over time 2) it illustrates the diversity such rights of way can support and their potential utility for conservation 3) we can identify win-win scenarios for management in powerline rights of way for conserving species diversity. Moreover, as we show, we have found new state records by sampling in these powerline rights of way, suggesting they have an as yet underappreciated value. We now expand on this objective on lines 110-116.***

2. I do not think the design is sufficiently robust to make conclusions on the effects of herbicide application on abundance and richness of insect communities. To address the second objective (“…compare the current management strategies to determine which has the potential to best support long-term pollinator diversity”), the abundance and diversity of flower-visiting insects are analyzed across 5 plots of 3 treatment types. I am sympathetic to the challenges inherent in large survey efforts of insects which often lead to smaller sample sizes due to trade-offs in how sampling is stratified across space and time. Nevertheless, the authors do not provide much description or context on the different plots beyond a categorical name and a liter/hectare amount of herbicide application. The significance of “basal” vs “foliar” is not explained (though I am assuming it has to do with how the herbicide was applied). Furthermore, although the study is cast as a comparison of management strategies, we are not provided background or context on these strategies in terms of how or why they are applied.

***It is true that we would rather have greater replication, but we feel that the long-term nature of the study sites makes them valuable for comparing management strategies. We now provide a greater description and context for the sites on lines 93-96 (and figure S1) and explain the basal vs foliar applications on lines 78-90.***

Below I have included some general comments by section.

Intro: The introduction is a good foundation yet very short and missing details which would enhance the rationale of the study. For example, more background context on powerline rights-of-way, how prevalent or widespread they are, the significance of early successional habitat for bees and other flower-visiting insects would enhance the study set-up. The addition of the research questions in this section would also improve the Introduction.

***We now provide more background context on how powerline rights of way are on lines 45-48, and the significance of early successional habitat for pollinating insects on lines 54-58. We also add the research questions on lines 66-70. We also use these research questions to guide the discussion in the revised ms.***

Materials and Methods: The first paragraph discusses treatment types by name, but it is not clear to me the significance of “high volume basal,” etc.

***We provide a more thorough description of the herbicide applications on lines 78-90.***

How were plots distributed within the ROW? Were they separated? Distances between them?

***We now provide more details about the spatial orientation of the plots on lines 93-96 and figure S1.***

Did you cover the same extent of each plot or walk transects?

Were multiple people collecting in the same plot at the same time?

How did you select which flowers to target for sampling?

Each plot was sampled 4 times May through August- was that once per month?

***We added more details of our collection events on lines 120-126, 131-133. We used a plot-based method where the collector traversed the area within the plot during the timed sample, collecting all visible insects contacting reproductive parts of flowers.***

More detail on the pan traps would be helpful, for example the purpose of multiple colors (for a non-entomological readership). Pan traps were placed between 2 plots- which ones and why? Why not place these within the net-sampled plots to potentially collect additional taxa? Line 97 states they were placed in an area with a similar background plant community, but it is not clear which plot or plot plant communities it is similar to.

***We provide a new supplemental figure to show the layout (Figure S1), including the position of the bowl sample, which was in the center of the sampling array. We include more details about the bowl sampling, including why we used multiple colors and why we chose not to conduct a bowl survey in each plot on lines 144-151.***

Results: In general, the Results section is hard to follow and does not flow well.

***We have substantially revised this section. In part, we added tables and removed parenthetical GLMM results from the text to improve the flow.***

Summary tables comparing richness and abundance values across plots and of the model results would be helpful additions to the manuscript.

***We added columns to table S1 with the bee and non-bee species richness and abundance for each plot (summed across all samples), and the herbicides used for each treatment. We also added three tables to the main text to summarize the GLMM results, and several additional supplemental tables with full model results.***

I do not think figure captions were included with the submission.

***We have now included figure captions at the end of the main text.***

Figure 2 is very confusing.

***We replaced this figure to make it more clear.***

Discussion: Some of the beginning Discussion information would be helpful in the Introduction. Highlight the main findings of the study early on in this section. Additional supporting literature would help put the results of this study in context of what others have found. The argument is made that ROW harbor diversity of these groups, but how do the richness and abundance values per plot compare to other studies? It can be difficult to compare across studies due to sampling effort, however the reader has no metric for assessing high diversity or abundance.

***We added some of the information from the discussion into the introduction. We also include some findings at the beginning, including comparing our collections to a study recently published by Kirkpatrick et al 2020, with a list of Pennsylvania bee species. We also discuss the total species richness in our ROW compared to another ROW study on lines 337-338.***

Reviewer #3: Review Russo et al. Power line ROWs and pollinator diversity

This study attempts to draw some general inferences about the diversity of bee and non-bee pollinators visiting flowers in managed experimental ROW plots. Although the experimental design has very little replication, and the explanatory variable (liters/hectare of applied herbicide) is not populated evenly across the range of variation, there are still valuable insights from this work that could be useful to other researchers.

***We would like to thank the reviewer for their helpful comments. We have addressed each comment on a point by point basis, demarcated by three asterisks (***). 

This paper could be strengthened by a analysis of the conservation value of each taxon (e.g. Natureserv G or S rankings) as another response variable to compare among the treatments. The interesting fact that several new distributional records were attained during this study speaks to knowledge gaps in the distribution of bee species in Pennsylvania; elaborating on these results in order to put these collections into regional or state context would improve the utility of this study. 126 bee species were observed, but how many were expected? This is a difficult question to answer and I’m not suggesting the authors should spend a great deal of time coming up with and justifying an expected species richness, but I am saying that a brief framing of these results in the context of the state and/or regional and local species pools would be useful pieces of information.

***Unfortunately, Natureserv did not include rankings for most of our species. In fact, only bumblebees have achieved rankings regarding their vulnerability. We did find some rare/threatened bumblebees in our study, however, by virtue of their rare nature, they were not in great enough abundance to statistically analyse their presence in the different treatments. We now refer to the species considered rare (Bombus fervidus, Bombus sandersoni, Macropis ciliata, Melissodes apicatus, Nomada xanthura, Heriades leavitti) in the results and discussion. We compare our bee species to the new checklist published by Kilkpatrick et al (2020) which catalogs the bee species of PA. Our study represents more than a quarter of the bee species of Pennsylvania in this small area. We also explain that we found six bee species considered rare by expert opinion, but do not see strong patterns in their distribution in the plots on lines 293-299, 393-397.***

Although the study infers that herbicide treatments have effects on the plant community, which then drive differences among the pollinator community visiting the flower assemblages during sampling periods, differences in the similarity and composition of the plant communities themselves are not explored. Plant species richness is proposed as a correlate of bee species richness, but the identity of plant species that are common and privately exclusive to the treatments is not discussed. Some mention of pollinator preference or specificity is given as an ad hoc hypothesis explaining results (line 250) but these data could be used to test such a hypothesis.

***We now include an analysis of the plant communities in the different plots…first, we include the number of plant species unique to each site in table S1 (this parallels plant species richness). Then, we describe the plant species that are present in all sites on lines 222-241, including a description of the specialist bees and their orientation in the plots. This mostly seems to agree with the host plant distribution (e.g. Asteraceae specialists were found where our botanical surveys recorded the most Asteraceae.***

This testable hypothesis is that (line 250) is that rare bee species might be more sensitive to increasing herbicide applications because herbicides may remove less common plant species. As mentioned above, no analysis of plant assemblages was provided but this hypothesis could at least be supported indirectly by evidence that plant species rare at low volumes herbicide applications are the species that are dropping out of the assemblage at higher volumes.

***It is hard to know which plant species are independently less common outside of our plots. We now discuss in detail which sites have the most unique/rare species (lines 293-299) and include a table of the presence/absence of the different plant species at the different sites in the supporting information.***

Another testable hypothesis, which is raised (line 171) but not tested, is that the significant difference in LVB bee abundance (wrt LVF and HVF) is a function of Bombus impatiens. A simple way to demonstrate that this species is driving this pattern is to remove this species from the analysis and compare abundance among plots.

***We reran the analyses after removing the two most abundant species (Bombus impatiens and Apis mellifera), and found no differences between the treatments, suggesting the most abundant species were, in fact driving the response. Otherwise, the abundance of species in the treatment sites is remarkably evenly distributed. We have added in this detail on lines 171-174, 262-269 and supporting information.***

Below I have some specific comments given with line numbers.

Line 98-100: The bees from the bowl surveys are presented without analysis, but it would be very interesting to know if these bees were more rare (higher imperilment ranking) than bees from the treatment plots. Additionally, the above comments about putting these collections into context with some treatment of the regional or state level fauna would really add meaning to these numbers.

***Most bee species do not have imperilment rankings, although we have some species considered more rare, or that are state records. We added a description of these rare species in the results and discussion. On the other hand, it is not appropriate to compare the bee bowl abundances to the netting because these different trapping methods can attract very different types of bees (lines 148-151).***

Lines 143- 147: See above, discussion of particular taxa devoid of ancillary data on species conservation value or life history could be enhanced

***We added some information about the most abundant bee species here (Bombus impatiens and Apis mellifera). We also now compare our collections to the newly published state list of PA bees. We found nearly 30% of the bee species of PA in our powerline ROWs, including roughly 35% of the non-native bee species.***

Line 157: This result cites Figure 1 incorrectly (bee abundance and richness correlation is given in Table S4 but not Figure 1)

***Thank you, we have corrected this reference.***

Line 163: There are several instances (see also 174 and 181) of the claim “significantly lower AIC”, this is misleading at best. The use of “significant” here may imply to a naïve reader that there is a hypothesis test comparing the AIC values, but that would be incorrect. I recommend that this language be modified perhaps as “Models including the categorical treatments as a fixed effect were more parsimonious than the continuous liters/ha herbicide usage as a fixed effect models”.

***We have adjusted the language as suggested.***

Line 167: A table of parameter estimates, confidence intervals and AIC scores would be very helpful to guide the reader through this section.

***We now have added three tables in the main text with details of the GLMMs, and supplemental tables with non-bee and the Bombus/Apis removal tables.***

Line 176: cite of Figure 2 at end of Line 178 is in the wrong place, should be cited at the end of statistics in Line 176

***We moved these stats to tables.***

Line 181: Table S6 indicates that the confidence intervals for LVF overlap with each of the other treatments.

***We removed this sentence.***

Line 189: Again the discussion of taxonomic diversity would be greatly enhanced with some analysis or representation of the local and regional species pool. Data on how many species were collected are hard to place into the proper context without some idea of how many bees (or non-bees) would be expected from such surveys, or how many exist in the region as a hole.

***There are no existing regional surveys of flower-visiting non-bee species, and in fact, it would be challenging to do so as there is a broad range of taxa that depend to varying degrees on floral resources. We can use GBIF records to get records of insect species in the state of Pennsylvania, but this includes a vast diversity of insects that never visit flowers at all, and there is no way to know which of these may be flower visitors. For example, GBIF has records of over 2500 beetle species in Pennsylvania, while our study found only 7 beetle species. However, the GBIF records include families of beetles that are carrion feeders, decomposers, predators, etc. which would likely not visit flowers. For most groups of insects, there is no way to taxonomically restrict the list to only flower visitors. The clearest group of flower-visitors we could find was the Syrphidae (hoverflies). GBIF lists 165 Syrphid species for Pennsylvania, while we found 13. We discuss this on lines 309-313.***

Line 238: the phrasing here is awkward and might be improved by striking “bee community more changeable over time” and replacing with something like “the phenological trajectory is stronger than treatment effects.

***Changed as suggested.***

Line 244- 247: Perfect opportunity to discuss ordination and analysis of plant assemblage data.

***We added discussions of the plant species richness on lines 222-241, 391-400.***

Line 249: This passage is unclear, whether singletons are singletons from the ENTIRE dataset of observations or singletons from a SITE or treatment. How many singletons were in each, and total?

***We added a sentence to clarify this on line 397-398.***

---

## [Decision Letter · Decision Letter 1]

23 Nov 2020

PONE-D-20-14623R1

Powerline right-of-way management and flower-visiting insects: How vegetation management can promote pollinator diversity

PLOS ONE

Dear Dr. Russo

Thank you for your revised submission, and I hope that you'll accept my deepest apologies for the delayed decision. Two reviewers were solicited to review, both agreed, and both provided timely, insightful reviews. However I had a number of challenges that delayed my recommendation. I hope you can appreciate that these are unusual times and will grant me some latitude. Nevertheless, the reviewers are in agreement and I concur that the revisions made have substantially improved the manuscript, and I commend you and your coauthors for your diligence. Both reviewers identify a few, largely cosmetic issues remaining in this manuscript. If you would consider those recommendations, resubmit with a response detailing changes made and/or justification/clarification for rejected recommendations, I will move to accept this manuscript without further peer review.

We look forward to receiving your revised manuscript.

Kind regards,

Mark A. Davis, Ph.D.

Academic Editor

PLOS ONE

Reviewers' comments:

Reviewer's Responses to Questions

**Comments to the Author**

1. If the authors have adequately addressed your comments raised in a previous round of review and you feel that this manuscript is now acceptable for publication, you may indicate that here to bypass the “Comments to the Author” section, enter your conflict of interest statement in the “Confidential to Editor” section, and submit your "Accept" recommendation.

Reviewer #1: (No Response)

Reviewer #2: All comments have been addressed

2. Is the manuscript technically sound, and do the data support the conclusions?

Reviewer #1: Yes

Reviewer #2: Yes

3. Has the statistical analysis been performed appropriately and rigorously? 

Reviewer #1: Yes

Reviewer #2: Yes

4. Have the authors made all data underlying the findings in their manuscript fully available?

Reviewer #1: Yes

Reviewer #2: Yes

5. Is the manuscript presented in an intelligible fashion and written in standard English?

Reviewer #1: Yes

Reviewer #2: Yes

6. Review Comments to the Author

Reviewer #1: I want to thank the authors for addressing each of my previous comments in their revision. I also appreciate the inclusion of the map – it really helped me understand the layout of this study. I feel strongly that these findings are important to publish. Realistic management recommendations from ecologists for the preservation of insect diversity are few and far between.

My primary comment relates to the layout of Fig. 2, I am a bit confused. Is the legend indicating that F2 had a greater herbicide usage than SF2, or are they only ranked by category? Is there a reason that Fig 1 groups each site within the category (LVB, LVF, HVF) but Fig 2 separates each site?

Very minor comments follow --

LN184 – I think this should be Simpson, not Simpson’s

LN339-340 – Near identical sentence used in introduction (LN45-47). (“…traverse a wide array of habitats and landscapes”)

Several sections could be rewritten to be clearer and potentially more succinct – LN296-299 (perhaps refer to supplementary data or include in treatment in species list LN294-295) and LN321-324

LN410-411 – Pollen predators is not a term I have seen used before – I assume it means pollen-feeding insects, but was not sure if it somehow included sit-and-wait predators.

Reviewer #2: Russo et al. address powerline Right-of-Ways (ROW) as habitat for flower-visiting insects. They describe species richness and abundance of bees and non-bees in relation to increasing volume per hectare application of herbicide and specific application type across 5 experimental plots in a managed ROW. In addition to identifying new records from Pennsylvania, they captured approximately 29 percent of the known bee species in the state. Flower-visiting insects were abundant with high richness at low levels of herbicide application, however bee species richness was negatively affected by higher levels of herbicide application. The authors conclude powerline ROW can be important habitat for bees and other flower-visiting insects.

Thank you for the opportunity to review the resubmission of the manuscript. I have some minor comments and suggestions for the authors below.

Introduction

Nice job reorganizing this section and including background information on ROW.

Lines 66-70: Suggestion for slight change of wording from research questions to objective statements-

“Our research objectives were to 1) assess species richness of flower-visiting bee and non-bee insects in the Vegetation Research and Demonstration Project at SGL33, 2) determine the effect of increasing herbicide application on flower-visiting insect abundance and species richness within a powerline ROW, and 3) address how long-term vegetation management affects the distribution of flower-visiting insects in a powerline ROW.”

Methods

The additional detail on sampling methods and treatment types was very helpful.

The abbreviations for low volume basal (LVB), etc, could be introduced here as they are then used in the Results section.

Lines 102-116: I’m not sure reiterating objectives and questions here is necessary and these two paragraphs could be eliminated.

Lines 155-157: A few additional details on how plant diversity was measured and a reference could be included.

Results

This section is much easier to follow.

Figure S1. The map and schematic of the sampling plots helped me to visualize the study. Consider moving it to the main body of paper.

Figure 1 (and Figure S3). I know there are different schools of thought on this, however showing raw abundance data as opposed to the log-transformed values used in the analysis would be more meaningful to the reader. Displaying raw abundance data would also keep consistency with Figure 2, in which richness data were log-transformed for the GLMM analysis but graphed as raw richness values.

Figure 2. To make the figure less busy and emphasize the point, you might graph litres/ha herbicide on the x axis (more informative than the plot names) against bee species richness with a second panel bar graph of plant richness, then eliminate the color in the bars. I recognize showing the x axis as continuous data may be tricky to make work visually. Alternatively, perhaps add an arrow along the x axis denoting increasing litres/ha of herbicide application. Are the outliers discussed in the text?

Figure 3. The caption is out of order in the text. The y axis is confusing. Should the label be “Number of Species”?

Discussion

The intro paragraph of this section makes a nice point for the role of ROWs as habitat in the context of species richness. I think the authors could also briefly highlight the strength of their sampling regime of morning and afternoon surveys across the active season and the implications for richness and abundance measures of flower-visiting insects.

Lines 363-367: I think the patterns driven by these two species is interesting. B. impatiens and A. melifera are known to be highly abundant locally. Any suggestions for why this may have been the case in your study? Proximity to nest locations or floral density? Perhaps the presence of a particularly abundant floral resource in these plots at the time plant richness was surveyed? Artifact of sample size?

7. PLOS authors have the option to publish the peer review history of their article (what does this mean?). If published, this will include your full peer review and any attached files.

Reviewer #1: No

Reviewer #2: No

---

## [Author Response · Author response to Decision Letter 1]

21 Dec 2020

PONE-D-20-14623R1

Powerline right-of-way management and flower-visiting insects: How vegetation management can promote pollinator diversity

PLOS ONE

Dear Dr. Russo

Thank you for your revised submission, and I hope that you'll accept my deepest apologies for the delayed decision. Two reviewers were solicited to review, both agreed, and both provided timely, insightful reviews. However I had a number of challenges that delayed my recommendation. I hope you can appreciate that these are unusual times and will grant me some latitude. Nevertheless, the reviewers are in agreement and I concur that the revisions made have substantially improved the manuscript, and I commend you and your coauthors for your diligence. Both reviewers identify a few, largely cosmetic issues remaining in this manuscript. If you would consider those recommendations, resubmit with a response detailing changes made and/or justification/clarification for rejected recommendations, I will move to accept this manuscript without further peer review.

We look forward to receiving your revised manuscript.

Kind regards,

Mark A. Davis, Ph.D.

Academic Editor

PLOS ONE

***Dear Dr. Davis, 

We would like to thank you and the anonymous reviewers for your helpful comments on this manuscript. We feel that it has been substantially improved by the reviewing process. We respond to the reviewer comments below on a point-by-point basis, demarcate by asterisks.***

Reviewers' comments:

Reviewer's Responses to Questions

Comments to the Author

1. If the authors have adequately addressed your comments raised in a previous round of review and you feel that this manuscript is now acceptable for publication, you may indicate that here to bypass the “Comments to the Author” section, enter your conflict of interest statement in the “Confidential to Editor” section, and submit your "Accept" recommendation.

Reviewer #1: (No Response)

Reviewer #2: All comments have been addressed

2. Is the manuscript technically sound, and do the data support the conclusions?

Reviewer #1: Yes

Reviewer #2: Yes

3. Has the statistical analysis been performed appropriately and rigorously? 

Reviewer #1: Yes

Reviewer #2: Yes

4. Have the authors made all data underlying the findings in their manuscript fully available?

Reviewer #1: Yes

Reviewer #2: Yes

5. Is the manuscript presented in an intelligible fashion and written in standard English?

Reviewer #1: Yes

Reviewer #2: Yes

6. Review Comments to the Author

Reviewer #1: I want to thank the authors for addressing each of my previous comments in their revision. I also appreciate the inclusion of the map – it really helped me understand the layout of this study. I feel strongly that these findings are important to publish. Realistic management recommendations from ecologists for the preservation of insect diversity are few and far between.

***Thank you very much for your helpful comments! We feel that your comments have substantially improved our manuscript. We have addressed your additional comments on a point by point basis below, demarcated by asterisks.***

My primary comment relates to the layout of Fig. 2, I am a bit confused. Is the legend indicating that F2 had a greater herbicide usage than SF2, or are they only ranked by category? Is there a reason that Fig 1 groups each site within the category (LVB, LVF, HVF) but Fig 2 separates each site?

***The litres/ha for each site is available in the supplemental table 1. The bars are arranged in increasing herbicide use order from left to right, so the highest litres per hectare is applied in F2 and the lowest in MH1. Figure 2 separates each site because the plant surveys were conducted at the site level, so we report the absolute value for plant species richness rather than an average across treatments for the purpose of this figure. We also made some edits to this figure to improve clarity, as suggested by reviewer 2 below.***

Very minor comments follow --

LN184 – I think this should be Simpson, not Simpson’s

***Changed as suggested.***

LN339-340 – Near identical sentence used in introduction (LN45-47). (“…traverse a wide array of habitats and landscapes”)

***We modified this sentence to reduce redundancy.***

Several sections could be rewritten to be clearer and potentially more succinct – LN296-299 (perhaps refer to supplementary data or include in treatment in species list LN294-295) and LN321-324

***We made these two sections more concise, as suggested.***

LN410-411 – Pollen predators is not a term I have seen used before – I assume it means pollen-feeding insects, but was not sure if it somehow included sit-and-wait predators.

***We clarified by saying “consumed pollen without acting as pollinators”..***

Reviewer #2: Russo et al. address powerline Right-of-Ways (ROW) as habitat for flower-visiting insects. They describe species richness and abundance of bees and non-bees in relation to increasing volume per hectare application of herbicide and specific application type across 5 experimental plots in a managed ROW. In addition to identifying new records from Pennsylvania, they captured approximately 29 percent of the known bee species in the state. Flower-visiting insects were abundant with high richness at low levels of herbicide application, however bee species richness was negatively affected by higher levels of herbicide application. The authors conclude powerline ROW can be important habitat for bees and other flower-visiting insects.

Thank you for the opportunity to review the resubmission of the manuscript. I have some minor comments and suggestions for the authors below.

***Thank you very much for your helpful comments. We feel they have substantially improved the quality of the manuscript. We address them on a point by point basis below, demarcated by asterisks.***

Introduction

Nice job reorganizing this section and including background information on ROW.

Lines 66-70: Suggestion for slight change of wording from research questions to objective statements-

“Our research objectives were to 1) assess species richness of flower-visiting bee and non-bee insects in the Vegetation Research and Demonstration Project at SGL33, 2) determine the effect of increasing herbicide application on flower-visiting insect abundance and species richness within a powerline ROW, and 3) address how long-term vegetation management affects the distribution of flower-visiting insects in a powerline ROW.”

***We changed this as suggested.***

Methods

The additional detail on sampling methods and treatment types was very helpful.

The abbreviations for low volume basal (LVB), etc, could be introduced here as they are then used in the Results section.

***We added the abbreviations as suggested.***

Lines 102-116: I’m not sure reiterating objectives and questions here is necessary and these two paragraphs could be eliminated.

***We deleted the first paragraph that reiterated the research question, but we included the second paragraph as an explanation for the motivation of establishing the baseline diversity because it was requested by a previous reviewer.***

Lines 155-157: A few additional details on how plant diversity was measured and a reference could be included.

***We have now included more details on the plant survey on lines 151-155 of the revised ms.***

Results

This section is much easier to follow.

Figure S1. The map and schematic of the sampling plots helped me to visualize the study. Consider moving it to the main body of paper.

***We have moved Figure S1 to the main text.***

Figure 1 (and Figure S3). I know there are different schools of thought on this, however showing raw abundance data as opposed to the log-transformed values used in the analysis would be more meaningful to the reader. Displaying raw abundance data would also keep consistency with Figure 2, in which richness data were log-transformed for the GLMM analysis but graphed as raw richness values.

***We changed these figures to represent the raw data.***

Figure 2. To make the figure less busy and emphasize the point, you might graph litres/ha herbicide on the x axis (more informative than the plot names) against bee species richness with a second panel bar graph of plant richness, then eliminate the color in the bars. I recognize showing the x axis as continuous data may be tricky to make work visually. Alternatively, perhaps add an arrow along the x axis denoting increasing litres/ha of herbicide application. Are the outliers discussed in the text?

***We now explain the outliers in the figure caption (also for figure 2 (previously figure 1)). We made this figure less busy and added a horizontal line to indicate increasing herbicide application on the x axis.***

Figure 3. The caption is out of order in the text. The y axis is confusing. Should the label be “Number of Species”?

***We corrected the order of the captions in the text. “Number of individuals” is correct. These graphs show the accumulation of new species as new individuals are collected. So, as we approach 1,000 specimens collected, we see a species richness (A) of over 100 species. We also include two other indices of species diversity, because evenness is also important to species diversity.***

Discussion

The intro paragraph of this section makes a nice point for the role of ROWs as habitat in the context of species richness. I think the authors could also briefly highlight the strength of their sampling regime of morning and afternoon surveys across the active season and the implications for richness and abundance measures of flower-visiting insects.

***We added a sentence to this effect on lines 341-342 of the revised ms.***

Lines 363-367: I think the patterns driven by these two species is interesting. B. impatiens and A. melifera are known to be highly abundant locally. Any suggestions for why this may have been the case in your study? Proximity to nest locations or floral density? Perhaps the presence of a particularly abundant floral resource in these plots at the time plant richness was surveyed? Artifact of sample size?

***We cannot truly know why these bees were more abundant in lower herbicide plots, however, we added a sentence on lines 358-359 that it might be due to the higher plant species richness in those plots.***

7. PLOS authors have the option to publish the peer review history of their article (what does this mean?). If published, this will include your full peer review and any attached files.

Do you want your identity to be public for this peer review? For information about this choice, including consent withdrawal, please see our Privacy Policy.

Reviewer #1: No

Reviewer #2: No

---

## [Editor Report · Decision Letter 2]

23 Dec 2020

Powerline right-of-way management and flower-visiting insects: How vegetation management can promote pollinator diversity

PONE-D-20-14623R2

Dear Dr. Russo,

We’re pleased to inform you that your manuscript has been judged scientifically suitable for publication and will be formally accepted for publication once it meets all outstanding technical requirements.

Kind regards,

Mark A. Davis, Ph.D.

Academic Editor

PLOS ONE
---

## [Editor Report · Acceptance letter]

28 Dec 2020

PONE-D-20-14623R2 

Powerline right-of-way management and flower-visiting insects: How vegetation management can promote pollinator diversity 

Dear Dr. Russo:

I'm pleased to inform you that your manuscript has been deemed suitable for publication in PLOS ONE. Congratulations! Your manuscript is now with our production department. 

Kind regards, 

on behalf of

Dr. Mark A. Davis 

Academic Editor

PLOS ONE